

# A decision tree model to predict liver cirrhosis in hepatocellular carcinoma patients: a retrospective study

Zheyu Zhou[1,*], Chaobo Chen[2,3,*], Meiling Sun[3], Xiaoliang Xu[4], Yang Liu[3], Qiaoyu Liu[4], Jincheng Wang[4], Yin Yin[4] and Beicheng Sun[1,4]

[1] Department of General Surgery, Nanjing Drum Tower Hospital, Chinese Academy of Medical Sciences & Peking Union Medical College, Graduate School of Peking Union Medical College, Nanjing, China
[2] Department of General Surgery, Xishan People's Hospital of Wuxi City, Wuxi, China
[3] Department of Hepatobiliary and Transplantation Surgery, The Affiliated Drum Tower Hospital of Nanjing University Medical School, Nanjing, China
[4] Department of General Surgery, The First Affiliated Hospital of Anhui Medical University, Hefei, China
* These authors contributed equally to this work.

## ABSTRACT

**Background:** The severity of liver cirrhosis in hepatocellular carcinoma (HCC) patients is essential for determining the scope of surgical resection. It also affects the long-term efficacy of systemic anti-tumor therapy and transcatheter arterial chemoembolization (TACE). Non-invasive tools, including aspartate aminotransferase to platelet ratio index (APRI), fibrosis-4 (FIB-4), and γ-glutamyl transferase to platelet ratio (GPR), are less accurate in predicting cirrhosis in HCC patients. We aimed to build a novel decision tree model to improve diagnostic accuracy of liver cirrhosis.

**Patients and Methods:** The Mann-Whitney U test, $\chi^2$ test, and multivariate logistic regression analysis were used to identify independent cirrhosis predictors. A decision tree model was developed using machine learning algorithms in a training cohort of 141 HCC patients. Internal validation was conducted in 99 HCC patients. The diagnostic accuracy and calibration of the established model were evaluated using receiver operating characteristic (ROC) and calibration curves, respectively.

**Results:** Sex and platelet count were identified as independent cirrhosis predictors. A decision tree model integrating imaging-reported cirrhosis, APRI, FIB-4, and GPR was established. The novel model had an excellent diagnostic performance in the training and validation cohorts, with area under the curve (AUC) values of 0.853 and 0.817, respectively. Calibration curves and the Hosmer-Lemeshow test showed good calibration of the novel model. The decision curve analysis (DCA) indicated that the decision tree model could provide a larger net benefit to predict liver cirrhosis.

**Conclusion:** Our developed decision tree model could successfully predict liver cirrhosis in HCC patients, which may be helpful in clinical decision-making.

Corresponding authors
Yin Yin, jyinjyin@sina.com
Beicheng Sun, sunbc@ahmu.edu.cn

## INTRODUCTION

Primary liver cancer is one of the most prevalent forms of cancer in the world (*Qiu et al., 2022*; *Sauzeau et al., 2022*; *Xu et al., 2022*). It is the fourth most common malignant tumor and the second leading cause of cancer death in China. Hepatocellular carcinoma (HCC) accounts for 75–85% of primary liver cancer (*Craig et al., 2020*; *Sung et al., 2021*). Liver cirrhosis caused by excessive drinking, obesity, hepatitis B virus (HBV) infection, or hepatitis C virus (HCV) infection is an essential contributing factor towards the development of HCC (*Ginès et al., 2021*).

Studies have found that about 1–8% of cirrhotic patients develop HCC yearly (*Ioannou et al., 2007*). HCC incidence also correlates with increased portal pressure or liver stiffness values, which are measured by transient elastography (TE) (*Masuzaki et al., 2009*; *Ripoll et al., 2009*). Radical resection is a vital method for the long-term survival of HCC patients, and the liver reserve function should be carefully evaluated before surgery. Future liver remnant (FLR) of more than 40% is necessary for surgery in cirrhotic patients. Due to the stealthy onset of HCC, less than 30% of patients are suitable for surgical treatment at initial diagnosis (*Zhou et al., 2020*). Local therapy, immunotherapy, and targeted therapy are the main treatments for patients with advanced HCC (*Wang et al., 2022b*). Studies have shown that the degree of liver cirrhosis and liver function status affect the long-term efficacy of systemic anti-tumor therapy and transcatheter arterial chemoembolization (TACE) (*Lencioni et al., 2016*; *Roth et al., 2023*). Therefore, evaluating the level of liver cirrhosis in HCC patients is of great significance.

Liver biopsy (LB) remains the gold standard for cirrhosis assessment. However, the invasive method is expensive and risks serious complications such as bleeding and infection. Sampling error in LB can also lead to an unreliable diagnosis (*Okada et al., 2007*; *Patel & Sebastiani, 2020*). TE is a non-invasive technique used to evaluate liver cirrhosis, but its results are associated with many factors, such as elevated aminotransferase or bilirubin levels, ascites, and extrahepatic cholestasis (*Millonig et al., 2008*; *Yan et al., 2013*). This limits the clinical application of TE in HCC patients. Furthermore, non-invasive tools, including the Child-Pugh (C-P) scoring system, aspartate aminotransferase to platelet ratio index (APRI), and fibrosis-4 (FIB-4), are less accurate in predicting cirrhosis in HCC patients (*Roth et al., 2023*; *Xiao et al., 2016*).

Therefore, we conducted this study intending to establish a new non-invasive model to improve the diagnostic accuracy of cirrhosis in HCC patients.

## MATERIALS AND METHODS

### Patient selection

We retrospectively reviewed the medical data of HCC patients who underwent curative liver resection at Nanjing Drum Tower Hospital from January 2020 to December 2022. The institutional review board of The Affiliated Drum Tower Hospital of Nanjing University Medical School approved this retrospective study, and the requirement for written informed consent was waived. The STARD 2015 reporting guideline was followed.

Patients with the following conditions were excluded from this study: recurrent HCC; HCC combined with distant metastasis; HCC combined with other tumors; had received local or systemic anti-tumor therapy (targeted therapy or immunotherapy) before surgery; had no des-γ-carboxy prothrombin (DCP) data before surgery. Patients in 2022 were assigned to the training cohort, and patients from 2020 to 2021 were assigned to the validation cohort. Routine laboratory tests were obtained 1 week before surgery, including hepatitis B virus surface antigen (HBsAg), hepatitis C virus antibodies (HCVAb), alpha-fetoprotein (AFP), DCP, red blood cell distribution width (RDW), neutrophil (NE), lymphocyte (LYM), monocyte (M), platelet (PLT), alanine aminotransferase (ALT), aspartate aminotransferase (AST), γ-glutamyl transferase (GGT), total bilirubin (TB), albumin (ALB), C-reactive protein (CRP), prothrombin time (PT), and international normalized ratio (INR). The albumin-bilirubin (ALBI) score is a simple, evidence-based, and objective method of assessing liver function in HCC patients. The ALBI score = $0.66 \times$ log (TB (μmol/L)) $- 0.085 \times$ ALB (g/L), and HCC patients were divided into three grades (grade 1: $\leq-2.60$; grade 2: $>-2.60$ to $\leq-1.39$; grade 3: $>-1.39$) according to the scoring results (*Johnson et al., 2015*). The formulas used to calculate inflammatory markers are as follows: GLR = GGT (U/L)/LYM count ($10^9$/L), PNI = ALB (g/L) + $5 \times$ LYM count ($10^9$/L), ALRI = AST (U/L)/LYM count ($10^9$/L), ANRI = AST (U/L)/NE count ($10^9$/L), SII = PLT count ($10^9$/L) $\times$ NE count ($10^9$/L)/LYM count ($10^9$/L), NLR = NE count ($10^9$/L)/LYM count ($10^9$/L), PLR = PLT count ($10^9$/L)/LYM count ($10^9$/L), LMR = LYM count ($10^9$/L)/M count ($10^9$/L), SIRI = M count ($10^9$/L) $\times$ NE count ($10^9$/L)/LYM count ($10^9$/L) (*Mao et al., 2021*).

## Pathological examination

Liver resection specimens were analyzed histologically by two experienced pathologists, who were blinded to the patients' clinical data. Any discrepancy was resolved by their discussion. The liver fibrosis status at the tumor periphery was determined according to the Metavir scoring system. F4 was considered as cirrhosis, F ≥ 3 as advanced fibrosis, and F ≥ 2 as significant fibrosis (*Bedossa & Poynard, 1996*). Typical pathological section images of F grade 0 to grade 4 were shown in Fig. S1.

## Imaging examination

All patients underwent contrast-enhanced computed tomography (CT) or Gd-EOB-DTPA-enhanced magnetic resonance imaging (MRI) within 2 weeks before surgery. Imaging results of all patients were evaluated for cirrhosis by two experienced radiologists. The diagnostic criteria for liver cirrhosis are as follows: the liver has a rough outline, an abnormal proportion of liver lobes, or uneven liver density. Indirect signs of liver cirrhosis include esophageal and gastric varices, ascites, and splenomegaly. Figure S2 presented typical imaging images of liver cirrhosis. Additionally, the maximum tumor diameter was recorded as the mean of two radiologist measurements. The number of tumors was recorded as solitary or multiple (*Yue et al., 2022*). Any discrepancy was resolved by their discussion.

## Serum liver cirrhosis diagnostic model

Since PLT, ALT, AST, and GGT are easily obtained parameters, APRI, FIB-4, and γ-glutamyl transferase to platelet ratio (GPR) are commonly used non-invasive diagnostic models of liver fibrosis. The formulas are as follows: FIB-4 = age (years) × AST (U/L)/PLT count $(10^9/L)$ × ALT $(U/L)^{1/2}$; APRI = (AST (U/L)/ULN) × 100/PLT count $(10^9/L)$ (*Itakura et al., 2021*; *Lemoine et al., 2016*).

## Statistical analysis and model development

Continuous and categorical variables were compared using the Mann-Whitney U test and $\chi^2$ test, respectively. The subsequent forward conditional multivariate logistic regression analysis (liver cirrhosis status is a binary variable) included variables with $p < 0.10$ in the training cohort (*Katz, 2003*). CT/MRI-reported cirrhosis, independent cirrhosis predictors, APRI, FIB-4, and GPR, were included to establish a novel decision tree using R software (version 3.6.1; *R Core Team, 2019*) and the R package rpart (*Wilson et al., 2018*). Decision tree models are supervised machine learning algorithms widely used in classification and regression tasks due to their interpretability and simplicity. The receiver operating characteristic (ROC) curve and the area under the curve (AUC) value were used to evaluate the diagnostic accuracy of the established decision tree model. The calibration curve was used to evaluate the calibration of the decision tree model, and the Hosmer-Lemeshow test was performed simultaneously (*Dong et al., 2021*; *Li et al., 2022*). Finally, the decision curve analysis (DCA) demonstrated the net clinical benefit at the different thresholds.

# RESULTS

## Baseline characteristics of enrolled patients

A total of 240 HCC patients were included in this study. There were no significant differences in clinical characteristics between the training cohort (141 patients) and the validation cohort (99 patients) (Table 1). Eighty-three patients (34.5%) were diagnosed with liver cirrhosis by pathological examination (training: 33.3%, 47 of 141; validation: 36.3%, 36 of 99; $p = 0.680$). Figure S3 showed consistency between serum biomarkers (APRI, FIB-4, and GPR) and pathological results.

## Risk factors for liver cirrhosis

In the training cohort, except for CT/MRI-reported cirrhosis and three serum diagnostic models (APRI, FIB-4, and GPR), age (years), RDW, LYM, PLT, PT, INR, PNI, ANRI were all related to cirrhosis ($p < 0.05$) (Table 2). Subsequently, the multivariable logistic regression analysis identified sex and PLT count as independent cirrhosis predictive factors ($p < 0.05$) (Table 3 and Fig. S4).

## Development and validation of the decision tree model

Since PLT is included in the formulas of APRI, FIB-4, and GPR, we investigated the effect of sex on the diagnostic accuracy of existing models for cirrhosis. The results showed that sex had little effect on diagnostic accuracy (Table 4). In addition, neither imaging nor

| Table 1 Baseline characteristics. | | | |
|---|---|---|---|
| **Variables** | **Training (*n* = 141)** | **Validation (*n* = 99)** | ***p* value** |
| **Basic demographics** | | | |
| Age (years) | 57.0 (51.0–68.0) | 57.0 (50.5–67.0) | 0.594 |
| Sex | | | 0.322 |
|    Male | 110 | 83 | |
|    Female | 31 | 16 | |
| BMI (kg/m$^2$) | | | 0.689 |
|    <24 | 82 | 61 | |
|    ≥24 | 59 | 38 | |
| **Laboratory findings** | | | |
| HbsAg | | | 0.559 |
|    Positive | 100 | 74 | |
|    Negative | 41 | 25 | |
| HCVAb | | | 0.326 |
|    Positive | 4 | 6 | |
|    Negative | 137 | 93 | |
| ALBI grade | | | 0.159 |
|    1 | 91 | 73 | |
|    2 or 3 | 50 | 26 | |
| AFP (ng/mL) | 22.2 (4.3–139.6) | 88.4 (6.2–676.5) | 0.139 |
| DCP (mAU/mL) | 169.7 (44.4–1,915.3) | 369.4 (42.3–2,359.5) | 0.606 |
| RDW (%) | 12.9 (12.4–13.4) | 12.8 (12.4–13.5) | 0.710 |
| NE (10$^9$/L) | 3.1 (2.4–3.8) | 3.0 (2.3–3.6) | 0.723 |
| LYM (10$^9$/L) | 1.5 (1.1–1.8) | 1.4 (1.1–1.8) | 0.622 |
| M (10$^9$/L) | 0.4 (0.3–0.5) | 0.4 (0.3–0.5) | 0.859 |
| PLT (10$^9$/L) | 157.0 (110.0–201.0) | 149.0 (107.0–181.5) | 0.471 |
| ALT (U/L) | 25.4 (18.4–36.3) | 25.4 (18.3–36.2) | 0.437 |
| AST (U/L) | 27.0 (21.8–34.3) | 26.5 (20.9–37.5) | 0.253 |
| GGT (U/L) | 44.2 (28.6–74.7) | 45.4 (25.6–77.4) | 0.432 |
| TB (μmol/L) | 13.7 (9.8–16.9) | 12.9 (10.1–16.6) | 0.883 |
| ALB (g/L) | 40.0 (38.7–42.0) | 40.6 (39.2–42.5) | 0.575 |
| CRP (mg/L) | 3.5 (2.1–5.2) | 2.8 (2.1–4.8) | 0.762 |
| PT (s) | 11.3 (10.8–11.8) | 11.6 (11.2–11.9) | 0.083 |
| INR | | | 0.294 |
|    ≤1 | 79 | 48 | |
|    >1 | 62 | 51 | |
| **Serum fibrosis tests** | | | |
| APRI | 0.5 (0.3–0.7) | 0.5 (0.3–0.9) | 0.258 |
| FIB-4 | 2.2 (1.5–3.2) | 2.3 (1.5–3.5) | 0.456 |
| GPR | 0.3 (0.2–0.5) | 0.3 (0.2–0.6) | 0.413 |
| **Inflammatory markers** | | | |
| GLR | 28.9 (17.5–62.6) | 30.5 (19.1–56.2) | 0.352 |

(Continued)

| Table 1 (continued) | | | |
|---|---|---|---|
| Variables | Training (*n* = 141) | Validation (*n* = 99) | *p* value |
| PNI | 46.7 (43.6–49.7) | 46.6 (42.2–50.0) | 0.400 |
| ALRI | 23.1 (14.2–42.8) | 29.9 (17.0–63.2) | 0.055 |
| ANRI | 10.5 (7.1–16.9) | 11.6 (7.1–22.1) | 0.091 |
| SII | 336.4 (225.0–521.1) | 382.5 (237.2–543.6) | 0.419 |
| NLR | 2.3 (1.7–3.1) | 2.5 (1.8–3.5) | 0.046 |
| PLR | 115.3 (77.9–165.0) | 128.8 (86.4–303.8) | 0.188 |
| LMR | 4.0 (3.3–5.0) | 4.0 (3.0–5.0) | 0.580 |
| SIRI | 0.7 (0.5–1.1) | 0.8 (0.5–1.1) | 0.721 |
| **Pathology parameter** | | | |
| F | | | 0.680 |
| 0–3 | 94 | 63 | |
| 4 | 47 | 36 | |
| **Imaging features** | | | |
| Tumor number | | | 0.928 |
| Solitary | 116 | 81 | |
| Multiple | 25 | 18 | |
| Tumor size (cm) | | | 0.938 |
| ≤5 | 89 | 62 | |
| >5 | 52 | 37 | |
| Cirrhosis | | | 0.691 |
| Present | 56 | 42 | |
| Absent | 85 | 57 | |

**Note:**

BMI, body mass index; HbsAg, hepatitis B virus surface antigen; HCVAb, hepatitis C virus antibodies; ALBI, albumin-bilirubin; AFP, alpha-fetoprotein; DCP, des-γ-carboxy prothrombin; RDW, red blood cell distribution width; NE, neutrophil; LYM, lymphocyte; M, monocyte; PLT, platelet; ALT, alanine aminotransferase; AST, aspartate aminotransferase; GGT, γ-glutamyl transferase; TB, total bilirubin; ALB, albumin; CRP, C-reactive protein; PT, prothrombin time; INR, international normalized ratio; APRI, aspartate transaminase to platelet ratio index; FIB-4, fibrosis-4; GPR, γ-glutamyl transferase to platelet ratio; GLR, γ-glutamyl transferase to lymphocyte ratio; PNI, prognostic nutritional index; ALRI, aspartate aminotransferase to lymphocyte ratio index; ANRI, aspartate aminotransferase to neutrophil ratio index; SII, systemic immune-inflammation index; NLR, neutrophil to lymphocyte ratio; PLR, platelet to lymphocyte ratio; LMR, lymphocyte to monocyte ratio; SIRI, systemic inflammation response index.

existing models had an area under the receiver operating characteristic (AUROC) greater than 0.67 in the entire cohort. That is why we intended to establish a novel model to improve the diagnostic accuracy of liver cirrhosis in HCC patients. According to the algorithm, sex was not included in the decision tree. For instance, patients with CT/MRI-reported cirrhosis and FIB-4 ≥ 6.0 are predicted to have cirrhosis, while patients without CT/MRI-reported cirrhosis and APRI < 0.38 are expected not to have cirrhosis (Fig. 1). The decision tree model had excellent diagnostic performance in the training and validation cohorts, with AUROC values of 0.853 (95% CI [0.783–0.923]) and 0.817 (95% CI [0.733–0.902]), respectively (Figs. 2A and 2B). Both calibration curves showed good calibration of the decision tree, indicating that the cirrhosis predicted by the model was consistent with the actual cirrhosis occurrence. The *p*-values of the Hosmer-Lemeshow test were 0.678 and 0.462, respectively (Figs. 2C and 2D). DCA for the decision tree model,

**Table 2 Comparisons of clinical characteristics among the training cohort.**

| Variables | F 0–3 ($n = 94$) | F 4 ($n = 47$) | $p$ value |
|---|---|---|---|
| **Basic demographics** | | | |
| Age (years) | 60.5 (53.0–68.8) | 55.0 (50.0–63.5) | 0.013 |
| Sex | | | 0.054 |
|    Male | 78 | 32 | |
|    Female | 16 | 15 | |
| BMI (kg/m$^2$) | | | 0.718 |
|    <24 | 56 | 26 | |
|    ≥24 | 38 | 21 | |
| **Laboratory findings** | | | |
| HbsAg | | | 0.078 |
|    Positive | 62 | 38 | |
|    Negative | 32 | 9 | |
| HCVAb | | | 0.108 |
|    Positive | 1 | 3 | |
|    Negative | 93 | 44 | |
| ALBI grade | | | 0.062 |
|    1 | 66 | 25 | |
|    2 or 3 | 28 | 22 | |
| AFP (ng/mL) | 15.4 (3.1–136.9) | 41.7 (8.6–176.3) | 0.457 |
| DCP (mAU/mL) | 236.1 (86.7–2,074.2) | 95.7 (32.9–820.5) | 0.555 |
| RDW (%) | 12.8 (12.4–13.3) | 13.1 (12.4–13.5) | 0.032 |
| NE ($10^9$/L) | 3.2 (2.6–3.9) | 2.9 (2.1–3.6) | 0.094 |
| LYM ($10^9$/L) | 1.6 (1.2–2.0) | 1.4 (1.0–1.6) | 0.010 |
| M ($10^9$/L) | 0.4 (0.3–0.5) | 0.3 (0.2–0.5) | 0.055 |
| PLT ($10^9$/L) | 176.0 (135.3–208.8) | 121.0 (82.5–161.5) | <0.001 |
| ALT (U/L) | 25.4 (18.7–36.0) | 25.4 (17.8–35.9) | 0.660 |
| AST (U/L) | 25.1 (20.5–31.3) | 32.6 (25.2–40.0) | 0.057 |
| GGT (U/L) | 45.0 (28.0–68.1) | 38.0 (29.7–88.9) | 0.579 |
| TB (μmol/L) | 13.1 (8.9–16.9) | 14.9 (11.8–17.2) | 0.178 |
| ALB (g/L) | 40.5 (39.1–42.1) | 39.5 (38.0–41.8) | 0.135 |
| CRP (mg/L) | 3.6 (2.1–5.0) | 3.2 (2.0–7.9) | 0.126 |
| PT (s) | 11.1 (10.7–11.6) | 11.8 (11.2–12.5) | <0.001 |
| INR | | | <0.001 |
|    ≤1 | 63 | 16 | |
|    >1 | 31 | 31 | |
| **Serum fibrosis tests** | | | |
| APRI | 0.3 (0.3–0.6) | 0.6 (0.5–0.9) | <0.001 |
| FIB-4 | 1.9 (1.4–2.5) | 2.7 (1.8–4.1) | <0.001 |
| GPR | 0.3 (0.1–0.4) | 0.4 (0.2–0.7) | 0.023 |
| **Inflammatory markers** | | | |
| GLR | 26.9 (16.3–57.1) | 33.1 (21.5–74.3) | 0.157 |

(Continued)

| Variables | F 0–3 (n = 94) | F 4 (n = 47) | p value |
|---|---|---|---|
| PNI | 47.4 (43.8–50.3) | 45.2 (42.5–47.8) | 0.025 |
| ALRI | 19.7 (12.5–37.8) | 27.7 (21.7–51.3) | 0.207 |
| ANRI | 8.9 (6.6–14.4) | 12.5 (9.0–27.1) | 0.004 |
| SII | 396.1 (244.6–543.9) | 287.7 (161.5–420.9) | 0.052 |
| NLR | 2.3 (1.7–3.1) | 2.3 (1.7–3.1) | 0.889 |
| PLR | 123.1 (83.2–180.8) | 105.0 (75.0–163.1) | 0.229 |
| LMR | 4.3 (3.4–5.0) | 3.8 (3.3–5.2) | 0.597 |
| SIRI | 0.8 (0.6–1.1) | 0.7 (0.4–1.1) | 0.608 |
| **Imaging features** | | | |
| Tumor number | | | 0.245 |
|     Solitary | 80 | 36 | |
|     Multiple | 14 | 11 | |
| Tumor size (cm) | | | 0.268 |
|     ≤5 | 56 | 33 | |
|     >5 | 38 | 14 | |
| Cirrhosis | | | <0.001 |
|     Present | 26 | 30 | |
|     Absent | 68 | 17 | |

**Note:**

BMI, body mass index; HbsAg, hepatitis B virus surface antigen; HCVAb, hepatitis C virus antibodies; ALBI, albumin-bilirubin; AFP, alpha-fetoprotein; DCP, des-γ-carboxy prothrombin; RDW, red blood cell distribution width; NE, neutrophil; LYM, lymphocyte; M, monocyte; PLT, platelet; ALT, alanine aminotransferase; AST, aspartate aminotransferase; GGT, γ-glutamyl transferase; TB, total bilirubin; ALB, albumin; CRP, C-reactive protein; PT, prothrombin time; INR, international normalized ratio; APRI, aspartate transaminase to platelet ratio index; FIB-4, fibrosis-4; GPR, γ-glutamyl transferase to platelet ratio; GLR, γ-glutamyl transferase to lymphocyte ratio; PNI, prognostic nutritional index; ALRI, aspartate aminotransferase to lymphocyte ratio index; ANRI, aspartate aminotransferase to neutrophil ratio index; SII, systemic immune-inflammation index; NLR, neutrophil to lymphocyte ratio; PLR, platelet to lymphocyte ratio; LMR, lymphocyte to monocyte ratio; SIRI, systemic inflammation response index.

**Table 3 Independent cirrhosis predictive factors in HCC patients.**

| Variables | Multivariate analysis | | |
|---|---|---|---|
| | β | OR | p value |
| Age (years) | −0.041 | 0.960 (0.920–1.002) | 0.060 |
| Sex | 1.016 | 2.763 (1.007–7.577) | 0.048 |
| HbsAg | 0.221 | 1.246 (0.420–3.702) | 0.691 |
| ALBI grade | 0.479 | 1.614 (0.124–20.970) | 0.714 |
| RDW | 0.277 | 1.319 (0.827–2.104) | 0.244 |
| NE | 0.211 | 1.235 (0.727–2.097) | 0.434 |
| M | 0.133 | 1.142 (0.037–35.700) | 0.940 |
| PLT | −0.016 | 0.985 (0.971–0.999) | 0.032 |
| AST | 0.006 | 1.006 (0.970–1.044) | 0.734 |
| PT | 0.630 | 1.877 (0.221–5.925) | 0.564 |
| INR | −3.886 | 0.021 (0.000–0.999) | 0.726 |

| Table 3 (continued) | | | |
|---|---|---|---|
| Variables | Multivariate analysis | | |
| | β | OR | p value |
| PNI | 0.029 | 1.030 (0.841–1.260) | 0.776 |
| ANRI | 0.017 | 1.017 (0.959–1.078) | 0.575 |
| SII | 0.001 | 1.001 (0.998–1.004) | 0.507 |

Note:
HCC, hepatocellular carcinoma; ROC, receiver operating characteristic curve; OR, odds ratio; AUC, area under the curve; HbsAg, hepatitis B virus surface antigen; ALBI, albumin-bilirubin; RDW, red blood cell distribution width; NE, neutrophil; M, monocyte; PLT, platelet; AST, aspartate aminotransferase; PT, prothrombin time; INR, international normalized ratio; PNI, prognostic nutritional index; ANRI, aspartate aminotransferase to neutrophil ratio index; SII, systemic immune-inflammation index.

**Table 4 The AUROC and cutoff values of diagnostic tools for liver cirrhosis.**

| | | Entire cohort | | |
|---|---|---|---|---|
| Sex | | Male | Female | All |
| Imaging-reported cirrhosis | | 0.649 (0.564, 0.734) | 0.665 (0.507, 0.823) | 0.658 (0.584, 0.731) |
| APRI | AUROC | 0.650 (0.569, 0.731) | 0.645 (0.484, 0.805) | 0.653 (0.582, 0.724) |
| | Cutoff | 0.47 | 0.51 | 0.47 |
| FIB-4 | AUROC | 0.573 (0.485, 0.661) | 0.669 (0.513, 0.826) | 0.603 (0.527, 0.678) |
| | Cutoff | 2.39 | 2.55 | 2.39 |
| GPR | AUROC | 0.585 (0.501, 0.670) | 0.577 (0.409, 0.745) | 0.577 (0.502, 0.652) |
| | Cutoff | 0.47 | 0.23 | 0.43 |

Note:
AUROC, area under the receiver operating characteristic; APRI, aspartate transaminase to platelet ratio index; FIB-4, fibrosis-4; GPR, γ-glutamyl transferase to platelet ratio.

imaging-reported cirrhosis, APRI, FIB-4, and GPR were presented in Fig. 3, which indicated that the decision tree model provided a larger net benefit than the other four models.

## DISCUSSION

Cirrhosis, a late stage of chronic liver inflammation where regenerative cirrhotic nodules replace normal liver structure, can eventually lead to liver failure. About one million people die of liver cirrhosis worldwide annually, and one-third of cirrhotic patients will develop HCC (*Ginès et al., 2021*). Assessment of cirrhosis in patients with chronic liver disease (CLD) or HCC should be the cornerstone of treatment decisions and prognosis (*Patel & Sebastiani, 2020*; *Wang et al., 2022a*).

Radical resection is the best treatment option for HCC patients with Barcelona Clinical Liver Cancer (BCLC) stage 0 or A (*Orimo et al., 2022*). The criteria for assessing the safety of resection are as follows: FLR > 20–30% and indocyanine green (ICG)-R15 < 10% (patients with C-P class A and without liver cirrhosis); FLR > 40% (patients with liver cirrhosis); FLR > 50% and IGG-R15 of 10–20% (patients with liver cirrhosis) (*Zhao & Cai, 2021*). Preoperative evaluation of the liver cirrhosis status of patients is an important basis for determining the scope of surgical resection. For patients within the Milan criteria, liver

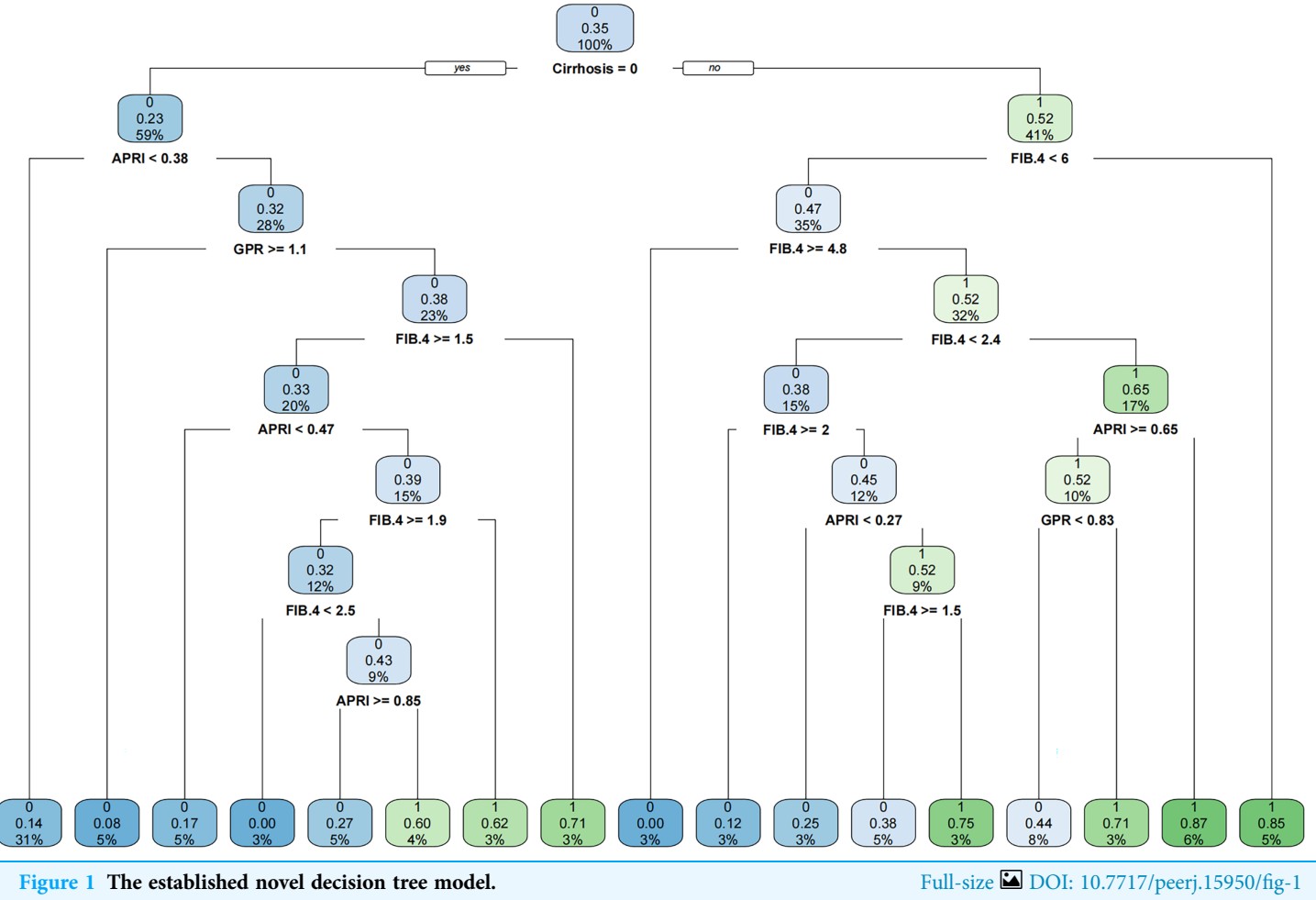

**Figure 1 The established novel decision tree model.**

transplantation (LT) is recommended as the first-line option for those unsuitable for liver resection (*Marrero et al., 2018*). Liver cirrhosis is a risk factor for postoperative recurrence of HCC (*Marasco et al., 2019*), and cirrhotic patients have a higher risk of developing HCC. Therefore, HCC patients with severe cirrhosis are ideal candidates for LT, accounting for 30–35% of European waiting list patients. Local and systemic treatments are options for HCC patients with BCLC stage B or C. The GIDEON study demonstrated that median overall survival (OS) was longer in C-P class A patients than in C-P class B patients (13.6 months *vs* 5.2 months) (*Marrero et al., 2016*). In addition, the prospective study by *Pressiani et al. (2013)* also showed that the OS and time to progression of C-P class A patients treated with sorafenib were longer than that of C-P class B patients. The C-P scoring system is the most widely used tool for assessing cirrhosis (*Roth et al., 2023*). Hence, assessing liver cirrhosis status in advanced HCC patients can predict treatment efficacy and prognosis. Compensated cirrhosis is asymptomatic, but it can progress to decompensated cirrhosis. Decompensated cirrhosis is associated with various complications (ascites, hepatic encephalopathy, variceal bleeding), leading to poor quality of life and frequent hospitalizations (*Ginès et al., 2021*). Thus, early identification of

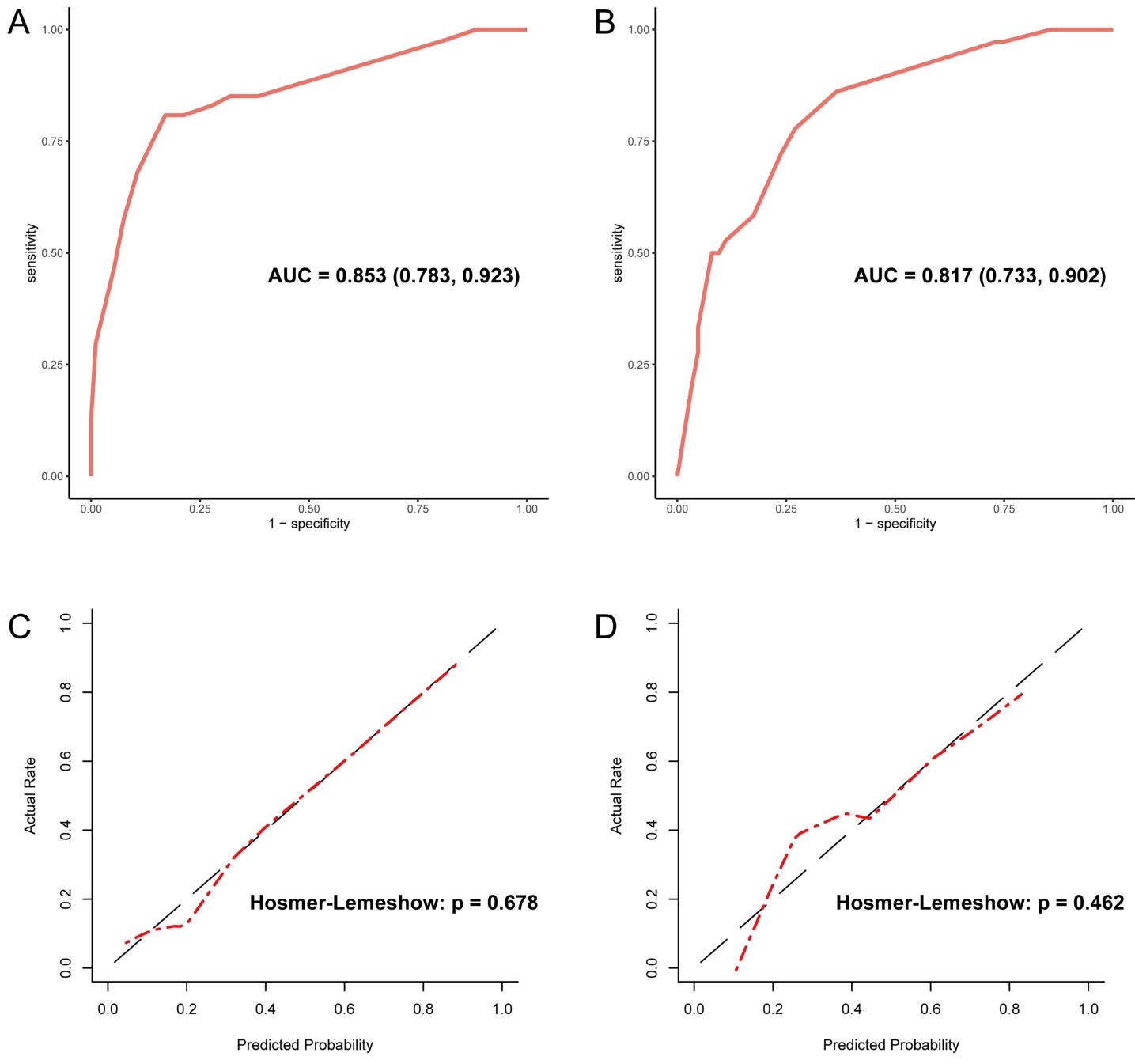

**Figure 2 Evaluation of diagnostic accuracy and calibration of the decision tree model.** (A) The ROC curve of the training cohort with an AUC value of 0.853. (B) The ROC curve of the validation cohort with an AUC value of 0.817. (C) The calibration curve of the training cohort. (D) The calibration curve of the validation cohort. ROC, receiver operating characteristic; AUC, area under the curve.

cirrhotic HCC patients for symptomatic and supportive treatment is imperative for mitigating the causative factors and preventing the progression of cirrhosis.

APRI, FIB-4, and GPR include five variables: age, PTL, ATL, AST, and GGT. Age is associated with cirrhosis (*Sterling et al., 2006*), and the incidence of HCC increases with age

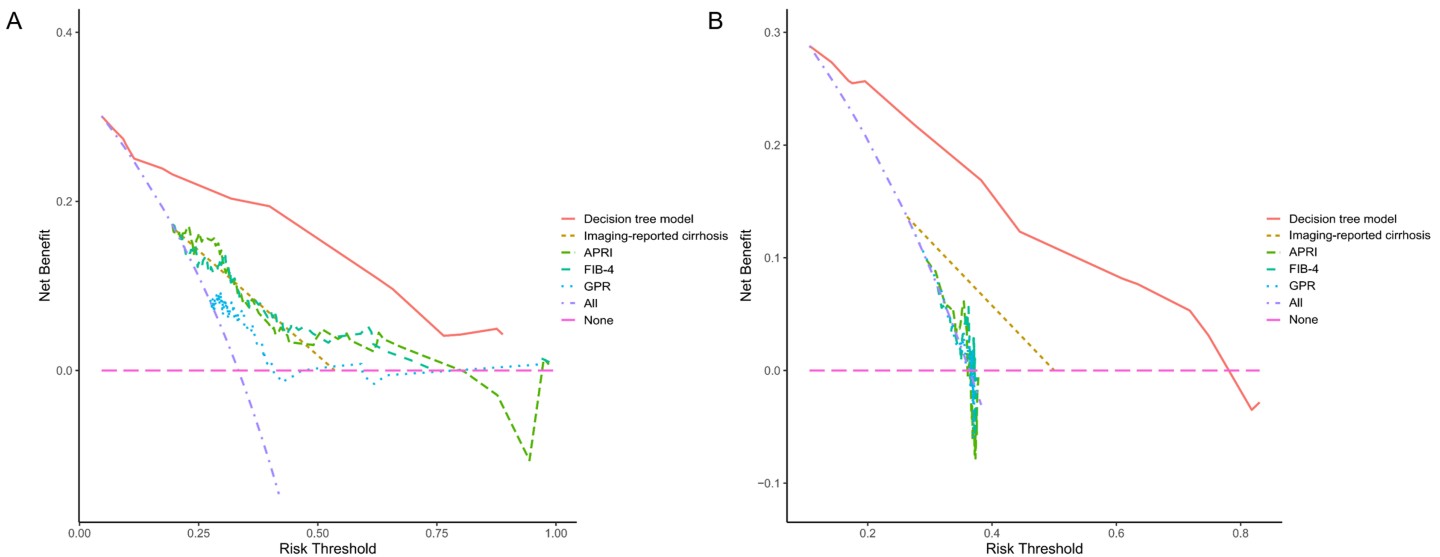

**Figure 3 The decision curve analysis for each model in the training (A) and validation (B) cohort.** The DCA indicated that the decision tree model could provide a larger net benefit to predict liver cirrhosis. DCA, decision curve analysis; APRI, aspartate transaminase to platelet ratio index; FIB-4, fibrosis-4; GPR, γ-glutamyl transferase to platelet ratio.

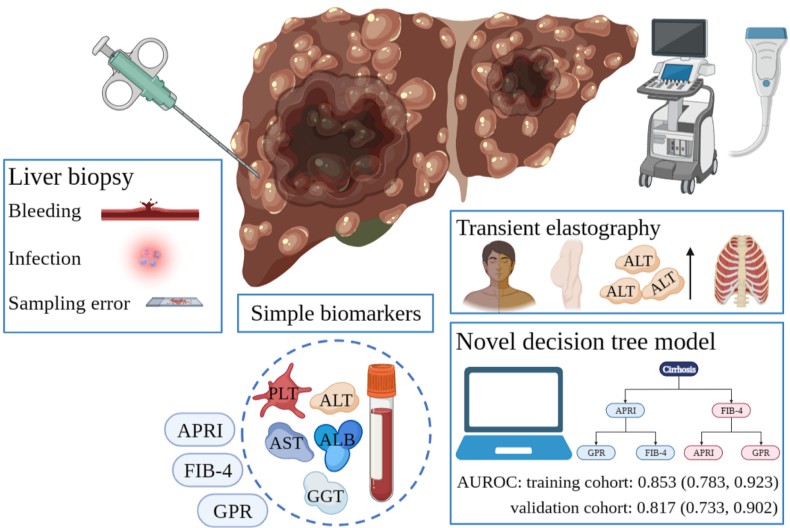

**Figure 4 Different diagnostic tools for liver cirrhosis in HCC patients.** LB is invasive and risks serious complications such as bleeding and infection. Sampling error in LB can lead to an unreliable diagnosis. Simple serum biomarkers are less accurate. Many factors affecting TE detection results limit its application in HCC patients. Our developed decision tree model can successfully predict liver cirrhosis. HCC, hepatocellular carcinoma; LB, liver biopsy; TE, transient elastography.

in all populations. Thrombocytopenia occurs in about 78% of patients with liver cirrhosis, and its degree of severity is an early prognostic marker. Decreased thrombopoietin (TPO) levels, hypersplenism secondary to portal hypertension, and myelosuppression induced by antiviral therapy may all lead to decreased PLT (*Peck-Radosavljevic, 2017*). Hepatocellular injury caused by chronic hepatitis can lead to elevated transaminases (especially AST) and

GGT (*Xiao et al., 2016*). To summarize, the variables included in serum diagnostic models correlate with liver fibrosis. The male-to-female ratio of HCC patients is about 2–2.5:1. However, this study found that being female is an independent risk factor for cirrhosis. This may be related to the sample size of the training cohort, which needs to be confirmed by further research.

A previous systematic review and meta-analysis showed that APRI and FIB-4 could identify HBV-related cirrhosis with moderate accuracy and sensitivity (*Xiao, Yang & Yan, 2015*). Meanwhile, *Xiao et al. (2016)* found that the AUC values for APRI and FIB-4 were 0.676 and 0.652 for diagnosing cirrhosis in HBV-related HCC patients. However, *Kim et al. (2016)* proposed that APRI and FIB-4 are unsuitable for evaluating the liver fibrosis of chronic hepatitis B (CHB) patients. This may be because the APRI and FIB-4 models were established in patients with HCV or HCV/HIV co-infection. In summary, the cause of liver cirrhosis and HCC is complicated, and the APRI and FIB-4 models lack high diagnostic accuracy in HCC patients. GPR was established by HBV patients in sub-Saharan African countries, where it was more accurate than APRI or FIB-4 (*Lemoine et al., 2016*). The study by *Lee et al. (2018)* reported that GPR was a valuable marker for assessing the degree of liver fibrosis in CHB patients (AUROC: 0.84; cut-off value: 0.299).

We focused on a noninvasive diagnostic model of cirrhosis status in HCC patients, and resected specimens from all patients were analyzed for liver fibrosis. APRI, FIB-4, and GPR are models established in CLD, and LB is the reference standard. LB specimens account for only 1/50,000 of the liver volume and may not represent the entire liver parenchyma (*Bravo, Sheth & Chopra, 2001*). In addition, LB specimen length (25 mm is considered optimal) and diameter (should be greater than 1 mm) also affect diagnostic accuracy (*Patel & Sebastiani, 2020*). Therefore, using surgical tissues for pathological examination is the advantage of our research. Like existing models, our decision tree model was also developed based on the binary classification of pathological examinations. This may not reflect the dynamic changes of liver fibrosis (*Patel & Sebastiani, 2020*). Although TE can identify liver stiffness in real-time, the Baveno IV consensus suggested that two tests on different days are required to confirm the validity (*de Franchis, 2015*), which will significantly increase the economic burden of HCC patients. Besides, the operator's inexperience is the main factor in the failure of TE detection (*Castéra et al., 2010*). Neither TE nor LB can be used as a routine detection method in areas with limited resources. Abnormal liver function, ascites, narrow intercostal space, obesity, and age over 52 years all affect TE detection (*Castéra et al., 2010*; *Millonig et al., 2008*; *Yan et al., 2013*).

Our research also has some shortcomings. First, selection bias brought about by retrospective research cannot be avoided. Second, there is a lack of external validation cohorts and prospective studies to further assess the model's accuracy. Third, the prevalence situation of HCC in China led us to include mainly HBV-associated patients. Finally, we did not build predictive models for significant and advanced liver fibrosis. This is because radiologists can only evaluate liver cirrhosis, and the diagnosis of cirrhosis is more clinically significant for HCC patients (*Wang et al., 2020*).

## CONCLUSIONS

In conclusion, the novel user-friendly decision tree model demonstrated excellent discrimination and calibration in predicting cirrhosis in HCC patients (Fig. 4). Since it is vital to know the cirrhosis status of HCC patients, the model can help clinicians in decision-making.

### Funding

This work was supported by the Anhui Provincial Key Research and Development Project (No. 202204295107020032), the National Natural Science Youth Foundation of China (No. 81902415, 82103135, and 82101850), and the Natural Science Youth Foundation of Jiangsu Province (No. BK20190116). The funders had no role in study design, data collection and analysis, decision to publish, or preparation of the manuscript.

### Grant Disclosures

The following grant information was disclosed by the authors:
Anhui Provincial Key Research and Development Project: 202204295107020032.
National Natural Science Youth Foundation of China: 81902415, 82103135, 82101850.
Natural Science Youth Foundation of Jiangsu Province: BK20190116.

### Competing Interests

The authors declare that they have no competing interests.

### Author Contributions

- Zheyu Zhou conceived and designed the experiments, performed the experiments, analyzed the data, prepared figures and/or tables, authored or reviewed drafts of the article, and approved the final draft.
- Chaobo Chen conceived and designed the experiments, performed the experiments, analyzed the data, prepared figures and/or tables, authored or reviewed drafts of the article, and approved the final draft.
- Meiling Sun conceived and designed the experiments, performed the experiments, analyzed the data, prepared figures and/or tables, authored or reviewed drafts of the article, and approved the final draft.
- Xiaoliang Xu performed the experiments, analyzed the data, prepared figures and/or tables, authored or reviewed drafts of the article, and approved the final draft.
- Yang Liu performed the experiments, analyzed the data, prepared figures and/or tables, authored or reviewed drafts of the article, and approved the final draft.
- Qiaoyu Liu performed the experiments, analyzed the data, prepared figures and/or tables, authored or reviewed drafts of the article, and approved the final draft.

- Jincheng Wang conceived and designed the experiments, performed the experiments, analyzed the data, prepared figures and/or tables, authored or reviewed drafts of the article, and approved the final draft.
- Yin Yin conceived and designed the experiments, authored or reviewed drafts of the article, and approved the final draft.
- Beicheng Sun conceived and designed the experiments, authored or reviewed drafts of the article, and approved the final draft.

## Human Ethics

The following information was supplied relating to ethical approvals (*i.e.*, approving body and any reference numbers):

The institutional review board of The Affiliated Drum Tower Hospital of Nanjing University Medical School approved this retrospective study.

## Data Availability

The raw data is available in the Supplemental Files.

## Supplemental Information

Supplemental information for this article can be found online at http://dx.doi.org/10.7717/peerj.15950#supplemental-information.

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
