# Peer review of "A decision tree model to predict liver cirrhosis in hepatocellular carcinoma patients: a retrospective study"

_PeerJ, doi:10.7717/peerj.15950_

## Round 0.1 · original submission · Minor Revisions

Please address the reviewers' comments.

Reviewer 1 ·

Basic reporting

Thanks for inviting me as the reviewer for this paper. The topic is interesting and not many studies investigated prediction of cirrhosis for HCC patients. I have suggestions as below:
1. I suggest the authors perform decision curve analysis (DCA) for each model (imaging-reported cirrhosis, APRI, FIB-4, GPR, and novel decision tree model) in the training and validation dataset. According to Table 4 and Fig. 2, although the novel decision tree model had a higher AUROC value, DCA should be performed to demonstrate the net clinical benefits.
2. In the pathological examination section: the liver fibrosis status was determined according to the Metavir scoring system. I suggest the authors provide typical pathological section images of F0, F1, F2, F3, and F4.
3. Since imaging-reported cirrhosis is an essential variable in the novel decision tree, I suggest the authors present typical imaging images of liver cirrhosis to identify their diagnostic criteria further.
4. The authors can reference Fig. 2 of this paper (PMID: 26109530). Box plots or violin plots can be used to show consistency between serum biomarkers (APRI, FIB-4, and GPR) and pathological results.
5. In this study, authors used random forest to establish the model. To the best of my knowledge, random forest is easier to cause the overfitted problem. If possible, authors can add external validation.

Experimental design

no comment

Validity of the findings

no comment

Additional comments

no comment

Reviewer 2 ·

Basic reporting

The authors developed a decision tree model for predicting cirrhosis in liver cancer patients, which compensates for the limitations of single forecasting indices and provides valuable guidance for clinical decision-making.
I read the study with great interest. The study is interesting and important. However, some important issues should be addressed.

Experimental design

1. The authors conducted a multivariate logistic regression to identify the risk factors associated with liver cirrhosis in patients with liver cancer, but did not specify the specific regression method used. It is recommended that further details be provided in the methodology section.
2. The authors employed differential clinical indicators for further analysis of cirrhosis risk factors between the cirrhotic and non-cirrhotic groups. What are the advantages of this approach compared to univariate logistic regression analysis?

Validity of the findings

1. Can authors provide forest plots for the multivariate logistic regression?
2. The author claims that except for FIB-4, sex had little effect on diagnostic accuracy in Table 4. How the author came to this conclusion.

Additional comments

There have been so many noninvasive methods for evaluating fibrosis. Authors should provide more detailed description about why the HCC patients should receive cirrhosis evaluation?

Reviewer 3 ·

Basic reporting

A very innovative report, the language is easy to understand.

Experimental design

Original primary research within Aims and Scope of the journal.

Validity of the findings

All underlying data have been provided; they are robust, statistically sound, & controlled.

Additional comments

Many thanks for inviting me to review titled "A decision tree model to predict liver cirrhosis in hepatocellular carcinoma patients"
A very innovative report, the language is easy to understand, and the overall research design is relatively reasonable.
There are still several issues that need to be further improved, so it may be better to explain this research report.
as follows:
a. Please add reference to sentence “Studies have found that about 1%-8% of cirrhotic patients develop HCC yearly” (line 51)
b. Are patients with HCC combined with distant metastasis also included? Please explain clearly.
c. Please explain why patients who had no des-γ-carboxy prothrombin (DCP) data before surgery were excluded from the study.
d. For HCC patients with liver cirrhosis, ICG-15 can also be used to evaluate the liver reserve function, which is related to the patient's surgical tolerance. Therefore, can it also be used as one of the non-invasive indicators for evaluating liver cirrhosis? This article does not involve the comparison of ICG-15, and is it routinely tested before surgery? Or explain in the discussion section.
e. Retrospective studies should follow some reporting guidelines, such as TRIPOD or STARD.

---

## Round 0.2 · accepted · Accept

Based on the Reviewer's comments, it is a good and careful revision.

Reviewer 1 ·

Basic reporting

no comment

Experimental design

no comment

Validity of the findings

no comment

Additional comments

This study has been revised and is recommended for publication

Reviewer 2 ·

Basic reporting

Well done!

Experimental design

No comments

Validity of the findings

No comments

Reviewer 3 ·

Basic reporting

no comment

Experimental design

no comment

Validity of the findings

no comment

Additional comments

no comment